# Comparative Behavior of Viscose-Based Supercapacitor Electrodes Activated by KOH, H_2_O, and CO_2_

**DOI:** 10.3390/nano12040677

**Published:** 2022-02-18

**Authors:** Stefan Breitenbach, Jiri Duchoslav, Andrei Ionut Mardare, Christoph Unterweger, David Stifter, Achim Walter Hassel, Christian Fürst

**Affiliations:** 1Wood K plus—Kompetenzzentrum Holz GmbH, Area Biobased Composites & Processes, 4040 Linz, Austria; c.unterweger@wood-kplus.at (C.U.); c.fuerst@wood-kplus.at (C.F.); 2Institute of Chemical Technology of Inorganic Materials (TIM), Johannes Kepler University Linz, 4040 Linz, Austria; achimwalter.hassel@jku.at; 3Center for Surface and Nanoanalytics (ZONA), Johannes Kepler University Linz, 4040 Linz, Austria; jiri.duchoslav@jku.at (J.D.); david.stifter@jku.at (D.S.)

**Keywords:** activated carbon, electrode materials, supercapacitor, viscose fibers, bio-based carbon, energy storage, activation agents, EDLC

## Abstract

Activated carbons derived from viscose fibers were prepared using potassium hydroxide, carbon dioxide, or water vapor as activation agents. The produced activated carbon fibers were analyzed via scanning electron microscopy and energy dispersive X-ray spectroscopy, and their porosity (specific surface area, total pore volume, and pore size distribution) was calculated employing physisorption experiments. Activated carbon fibers with a specific surface area of more than 2500 m^2^ g^−1^ were obtained by each of the three methods. Afterwards, the suitability of these materials as electrodes for electrochemical double-layer capacitors (supercapacitors) was investigated using cyclic voltammetry, galvanostatic measurements, and electrochemical impedance spectroscopy. By combining CO_2_ and H_2_O activation, activated carbon fibers of high purity and excellent electrochemical performance could be obtained. A specific capacitance per electrode of up to 180 F g^−1^ was found. In addition, an energy density per double-layer capacitor of 42 W h kg^−1^ was achieved. These results demonstrate the outstanding electrochemical properties of viscose-based activated carbon fibers for use as electrode materials in energy storage devices such as supercapacitors.

## 1. Introduction

One of the most pressing global challenges is reducing greenhouse gas emissions and fossil fuel consumption to combat global warming. This problem can be solved by energy production from renewable energy sources. Harvesting a power source as abundant as the sun presents certain advantages and it is exploited in photocatalysis and photovoltaics [1,2]. Many nations and global organizations have proposed plans to address the global warming challenge [3]. The EU, for example, intends to cut greenhouse gas emissions by at least 40% by 2030, while increasing energy from renewable sources to at least 32% [4]. Climate neutrality is to be reached by the year 2050, meaning that either no more greenhouse gases will be emitted, or their emissions will be entirely compensated [4]. The usage of renewable energy sources, on the other hand, creates additional challenges. The amount of energy generated by wind or solar energy, for example, fluctuates dramatically over time, posing significant problems for grid stability and power grid management. Energy storage systems are essential in order to utilize renewable energies reliably, sustainably, and consistently.

Supercapacitors are energy storage devices gaining popularity due to their ability to be charged and discharged very rapidly, needing little maintenance, and having a very high power density and a long life of over 1 million charge and discharge cycles [5,6,7,8]. Carbon materials are commonly used to fabricate electrodes for such supercapacitors because they are easy to process and are chemically and thermally stable. Moreover, their structural properties can be easily modified and optimized. Activated carbons are widely used electrode materials because they have a large specific surface area (1000–3000 m^2^ g^−1^) and a high total pore volume (0.5–2 cm^3^ g^−1^) at a reasonable cost (3.65 € kg^−1^) [9,10,11]. In the last two decades, the interest in sustainable carbon precursors increased drastically, and all sorts of biomass have been tested as precursors for activated carbon [12,13,14]. However, the availability and consistency of the material quality is always an issue that has to be considered when using natural materials [15]. Fibrous precursors are of special interest, as activated carbon fibers (ACFs) are a thrilling alternative to traditional granular and powdered activated carbons (ACs). The great advantage of ACFs is their low electrical resistance along the fiber axis and the excellent contact with the current collector [16,17]. For achieving the required carbon properties, a number of chemical and physical activation routes have been developed for increasing carbon surface areas and developing the pore network in carbon materials [18,19,20]. Despite this, fibrous activated carbon materials made from bio-based precursors have been poorly studied. This is partly due to the fact that only high-purity materials such as viscose or lignin fibers can be used due to the consistency of the material quality. Another reason for the lack of publications is the carbonization and activation process, which must be selected in such a way that the fiber structure is at least partially retained, even in the electrode. Based on these findings, in the current work, viscose fibers were chosen as precursor material. These are artificially-made fibers from bio-based resources, thus combining sustainability and availability in good and constant quality, while also offering the full advantage of fibrous materials. Three different activation methods are applied, and their effects on the porosity of the ACFs and their properties on the electrochemical performance of the supercapacitor are investigated. The physical activation after carbonization by means of carbon dioxide and water vapor is considered. In addition, the chemical activation by means of potassium hydroxide, which takes place during the carbonization step, is examined.

## 2. Materials and Methods

### 2.1. Preparation of the Activated Carbon Fibers

After drying for 12 h at 90 °C, viscose fibers (1.7 dtex, 38 mm) were treated with a 37.9 mmol L^–1^ DAHP solution for 15 min. The fibers were dried again (12 h at 90 °C) before being carbonized in nitrogen atmosphere in a chamber furnace (HTK8, Carbolite Gero GmbH, Neuhausen, Germany). The temperature program consisted of a heating ramp of 10 °C min^−1^ up to 850 °C and an isothermal step at this temperature for 30 min.

The activation with H_2_O and/or CO_2_ was done after carbonization in a rotary kiln (RSR-B 120/500/11, Nabertherm GmbH, Lilienthal, Germany). The furnace was heated to 870 °C under nitrogen and kept isothermal for 30 min prior to the actual activation. A gas flow of 80 L h^−1^ for 180 min was used for the CO_2_ activation. For the activation with H_2_O, 1.0 mL min^−1^ of water was injected with a peristaltic pump for 180 min.

Activation using KOH took place during the carbonization step. The literature-recommended approach of mixing the precursor with four times the amount of KOH and subsequent carbonization did not work, as the fiber structure should not be compromised due to grinding with KOH. [21]. Instead, the DAHP-impregnated and dried viscose fiber was impregnated again with a 50 wt.% KOH solution. The ACF made like this was washed several times with a 10% HCl solution and with distilled H_2_O until a neutral pH was achieved.

### 2.2. Electrode Fabrication and Electrochemical Characterization

The ACFs were ground in a mortar mill (RM 200, Retsch GmbH, Haan, Germany). After adding 10 wt.% graphite (C-nergy KS-L, Imerys, Paris, France) and 10 wt.% polytetrafluoroethylene (PTFE; Sigma Aldrich, St. Louis, MO, USA), the ACFs were ground again until a kneadable dough was obtained. This dough was rolled out to a thickness of 90–100 µm using a sheet metal roller, and electrodes were punched out. The electrodes were dried in vacuo overnight at 110 °C.

Two symmetrical electrodes soaked with a 1 M triethylmethylammonium tetrafluoroborate (TEMA BF_4_, TCI Deutschland GmbH, Eschborn, Germany) solution in propylene carbonate (PC, Acros Organics N.V., Geel, Belgium) were used in a Swagelok^®^-type 2-electrode test cell. To prevent a short circuit, a separator (Celgard^®^ 3401, Celgard LLC, Charlotte, NC, USA) was used between the electrodes. C-coated aluminum foil was employed as current collector (z-flo 2651, Coveris Management GmbH, Vienna, Austria). All cells were built under an argon atmosphere.

A potentiostat (Vertex.One, IviumTechnologies BV, Eindhoven, The Netherlands) was used for electrochemical impedance spectroscopy (EIS) measurements, cyclic voltammetry (CV), and galvanostatic discharge curves (GDC). EIS measurements were conducted with 10 mV of AC voltage amplitude in the frequency range of 1 MHz–0.01 Hz. The gravimetric capacitance *C*_S_ was calculated from CV and GDC data using Equations (1) and (2):(1)$$C_{S,\mathrm{CV}}=\frac{\int_{V_{i}}^{V_{f}}i\;\cdot \;dV}{2\;\cdot {\;m}_{E\;}\cdot \;v}$$
(2)$$C_{S,\mathrm{GDC}}=\frac{2\;\cdot \;i\;\cdot \;t}{m_{E\;}\cdot \;\Delta V}$$

The specific capacitance *C*_S,CV_ of one electrode was determined by the integral $\int_{V_{i}}^{V_{f}}i\;\cdot \;dV$ of the positive part of the 5th CV curve, limited by *V*_f_ and *V*_i_ of the cell voltage Δ*V*, scan rate *v*, and the mass of the active material *m*_E_ of one electrode. For the GDC measurements, *C*_S,GDC_ was calculated using the current *i*, the discharge time *t*, *m*_E_ and Δ*V*. The energy density *E*_S_ and power density *P*_S_ for a device were calculated using Equations (3) and (4):(3)$$E_{S}=\frac{1}{8}\;\cdot {\;C}_{S,\mathrm{GDC}}\;\cdot \;{\left(\Delta V\right)}^{2}$$
(4)$$P_{S}=\frac{E_{S}}{t}$$

### 2.3. Structural Characterization

A volumetric sorption analyzer (Autosorb-iQ, Anton Paar QuantaTec Inc, Boynton Beach, FL, USA) was used to determine the specific surface area, pore size distribution, and total pore volume of the ACFs by N_2_ isothermal adsorption (77 K). The samples were degassed for 2 h at 350 °C. The non-local density functional theory (NLDFT) and the Brunauer–Emmett–Teller (BET) method were used to compute the pore size distribution (PSD) and the specific surface area *S*_BET_ [22,23]. At a relative pressure of 0.99, the total pore volume *V*_tot_ was estimated to be equal to the liquid volume of the adsorbate (N_2_).

An Ultim Max 100 EDX detector (Oxford Instruments, Abington, United Kingdom) coupled to a scanning Auger electron spectroscopy microscope (JAMP-9500F, JEOL Ltd., Akishima, Japan) was used to characterize structural and elemental properties of the samples. Energy-dispersive X-ray spectroscopy (EDX) measurements were performed at an acceleration voltage of 5 kV. The system was also used to take scanning electron microscopy (SEM) images.

## 3. Results and Discussion

To achieve qualitative comparability, the activation conditions were chosen so that ACFs with *S*_BET_ of around 2500 m^2^ g^−1^ were produced. CO_2_ has been previously studied, and the prepared AFC yielded very good results in supercapacitor electrodes [24]. In the current study, activation by H_2_O using identical activation temperature and time yielded comparable ACFs. For KOH activation, a completely different approach had to be used. Here, the DAHP-impregnated and dried fiber was treated again with a 50 wt.% KOH solution. After drying, chemical activation took place simultaneously with the carbonization step. In physical activation with CO_2_ or H_2_O, carbon is partially gasified by the oxidant. The gasification of carbon by CO_2_ is dependent on the Boudouard reaction [25]. This reaction describes the equilibrium between CO_2_ and CO that is established during the reaction with carbon Equation (5). High temperatures shift the equilibrium to the product side (CO) due to the endothermic reaction. In this way, carbon in the sample was partially gasified.
C + CO_2_ ⇌ 2 CO(5)

Activation with H_2_O as the activating gas works according to the water gas equilibrium equation (Equation (5)), in which carbon and steam are in equilibrium with CO and hydrogen gas. A detailed description of this reaction, which can be very complex, can be found in the corresponding literature [25,26].
C + H_2_O ⇌ CO + H_2_(6)

With the use of KOH, the carbon skeleton was additionally intercalated and expanded by potassium [18]. All three tested activation techniques led to ACFs reaching the desired range of *S*_BET_ = 2500 m^2^ g^−1^ (Table 1). The ACF activated with CO_2_ had the highest *S*_BET_ of 2737 m^2^ g^−1^. The high *S*_BET_ attained here is not surprising, given that activation by CO_2_ has been extensively investigated, and the effects of DAHP as an impregnating agent have also been described previously [24,27]. H_2_O- and KOH-activated fibers showed a marginally lower *S*_BET_ of 2553 m^2^ g^−1^ and 2516 m^2^ g^−1^, respectively. With 1.35 cm^3^ g^−1^, the CO_2_-activated sample showed the highest *V*_tot_. The H_2_O-activated ACF had with 1.27 cm^3^ g^−1^ a higher *V*_tot_ than the KOH-activated sample (1.10 cm^3^ g^−1^).

The total yield calculated from the difference of the mass of dried viscose fiber and ACF divided by the mass of dried fiber is also shown in Table 1. The total yield of the KOH activation was the lowest at 2.1%. In addition to the formation of pores, this was due to the fact that a very fine powder was produced, which had to be washed several times. The yield of activation using CO_2_ was much higher at 9.2%. The highest total yield of 23.9% was achieved with activation using H_2_O. The total yield of ACFs activated with CO_2_ or H_2_O was composed of the partial yields of the carbonization step and the activation step [24]. The carbonization yield in both cases was 31%.

The PSD (Figure 1b) calculated from the adsorption isotherms (Figure 1a) using nitrogen at 77 K explains why the H_2_O-activated ACF had a higher *V*_tot_ at lower *S*_BET_ than the KOH-activated ACF. The AC produced by KOH activation had pores with a diameter *d*_P_ < 0.7 nm and in the range of *d*_P_ = 1.0–2.5 nm. On the other hand, activation with CO_2_ or H_2_O hardly led to pores smaller than 0.9 nm. The majority of the pores had a diameter of *d*_P_ = 0.9–3.5 nm. The CO_2_-activated sample had a larger pore volume (around 2.5 nm) than the water-activated ACF. The H_2_O-activated fiber was the only sample to show pores with a diameter around 3.7 nm.

The CVs of the EDLCs were recorded at a scan rate of 10 mV s^−1^ with a potential window of −2.7–2.7 V (Figure 2a). Decomposition of the electrolyte did not occur at a potential window of ±2.7 V, as was already demonstrated [28,29,30]. The test cell prepared from the CO_2_- and the H_2_O-activated fibers had a quasi-rectangular shape, which is typical for EDLCs. The KOH-activated sample showed a parallelogram, indicating a higher internal resistance [31]. A similar trend could be seen in the GDCs measured at 1.0 A g^−1^ (Figure 2b). While the EDLC made from CO_2_-activated ACFs showed a relatively small internal resistance (IR) drop, the other two samples had a much stronger decline. The calculated equivalent series resistance (ESR) was 584 Ω for the KOH-activated sample, 430 Ω for the H_2_O-activated sample, and only 254 Ω for the CO_2_-activated EDLC. The specific capacitance *C*_S,CV_, derived from the CVs at various scan rates *v*, is shown in Figure 2c. At the smallest scan rate studied (1 mV s^–1^), the highest specific capacitance was achieved. The H_2_O-activated sample showed the largest specific capacitance at 180 F g^−1^, the electrode made of CO_2_-activated fibers 171 F g^−1^, and the KOH-activated sample 166 F g^−1^. This trend persisted at more rapid scan rates. Even at very fast scan rates of 300 mV s^−1^, the investigated EDLCs still reached values of 45 F g^−1^ (H_2_O), 34 F g^−1^ (CO_2_), and 24 F g^−1^ (KOH). Evaluation of specific capacitance by GDC is recommended over that by CV because side reactions can occur, especially at the edges of the potential window, which can enlarge the specific capacitance calculated by CV [32]. The calculated specific capacitance *C*_S,GDC_ of the samples (Figure 2d) behaved slightly differently than that of *C*_S,CV_. Here, the KOH-activated sample showed the maximum specific capacitance of 140 F g^−1^ at the lowest specific current *I*_S_ of 0.1 A g^−1^. For the electrodes produced from CO_2_-activated ACFs, *C*_S,GDC_ was 121 F g^−1^, and for the H_2_O-activated fibers, 112 F g^−1^. With the increase of *I*_S_, a larger voltage drop occurred, probably caused by the high internal resistance of the KOH sample, causing *C*_S,GDC_ to drop sharply. At currents of 2.0 A g^−1^ and higher, the IR drop was so large that no specific capacitance could be determined. The best performance at discharge currents above 0.6 A g^–1^ was shown by the CO_2_-activated sample. From the GDCs, the specific energy *E*_S_ and the specific power *P*_S_ of the complete EDLC (not per electrode) were calculated and are shown in a Ragone plot (Figure 2e). The highest power of 1360 W kg^−1^ was obtained with the CO_2_-activated electrodes, followed by the H_2_O-activated (626 W kg^−1^) and the KOH-activated electrodes (410 W kg^−1^). Additionally, the highest specific energy of 25.9 W h kg^−1^ was achieved by the CO_2_-activated fibers. The other two samples reached specific energies of 20.8 W h kg^−1^ (H_2_O) and 18.9 W h kg^−1^ (KOH) per device, respectively.

The morphology of the electrodes was examined by SEM (Figure 3). The electrodes from the CO_2_- and H_2_O-activated ACFs displayed a very similar morphology, including fibrous parts. The electrode made from the KOH-activated sample showed hardly any clear fibrous features. Only granular pieces and fibrous fragments could be found. This was already evident after activation. While the fiber structure was completely preserved during activation with CO_2_ or H_2_O, KOH activation resulted in a powder. During activation with KOH, metallic potassium intercalated into the carbon lattice and expanded it [18,33]. This expansion seemed to destroy the fibrous structure of the sample.

EDX spectra (Table 2) were used to examine the effect of the activation method on the chemical composition. The electrode activated by KOH had only oxygen as a heteroatom with a content of 2.6 at.% and therefore a high carbon content of 97.4 at.%. The multiple washing steps of the AC with HCl and H_2_O probably resulted in this low amount of heteroatoms despite impregnation. It must be noted here that neither the CO_2_-activated nor the H_2_O-activated fibers were washed after activation. The influence of the impregnation using DAHP was clearly detectable on the electrode made from CO_2_-activated fibers. The carbon content was the lowest at 88.3 at.%, and in addition, oxygen (8.0 at.%), phosphorus (2.9 at.%), and nitrogen (0.9 at.%) could be found. In contrast, the electrode from the H_2_O-activated fibers was very pure. The carbon content was very high at 98.6 at.%, with only 1.4 at.% oxygen. Due to otherwise identical conditions, it can be concluded that the heteroatoms were washed out during H_2_O activation. 

In addition to the previous electrochemical measurements, the EDLCs were characterized using EIS, and the data were fitted to an equivalent circuit (Figure 4). The resistances R1 and R2 of the test cells calculated by fitting are shown in Figure 4c. The KOH-activated sample had the highest serial resistance R1 of 265 mΩ. The reason for this may be linked to the morphology of the electrode showing an increased grain boundary area/volume (see Figure 3c). While the other two electrodes consisted at least partly of fibers, along whose axis the electrical resistivity was low, the KOH electrode consisted of almost granular particles with many interfaces, whereby the resistance was higher. For the charge-transfer resistance R2, the picture was different. In contrast to the H_2_O-activated sample (R2 = 39 mΩ), the CO_2_-activated sample showed a relatively high charge-transfer resistance R2 of 370 mΩ. These findings contradict Ma et al., who found that P-doping decreases the charge-transfer resistance via improving the electrode’s wettability [34]. The charge-transfer resistance strongly depends on the accessibility of the pores by the electrolyte [16]. The pores in the H_2_O-activated sample with a diameter around 3.7 nm (see Figure 1b) could lead to better accessibility, which could outweigh the effect of P-doping.

Since a longer activation time with CO_2_ resulted in very poor yield, the specific surface area of the ACFs could not be further increased using CO_2_. Following the finding that H_2_O activation also achieved very good results, an attempt was made to first activate the carbonized fiber impregnated with DAHP using CO_2_ and then to further increase the surface area using H_2_O. The parameters for the CO_2_ activation were set to be identical to those of the previous CO_2_ activation, followed by 1 h of H_2_O activation with 1 mL min^−1^ at 870 °C.

The N_2_ isotherms and the PSDs derived are displayed in Figure 5. By H_2_O activation following the CO_2_ activation, the specific surface area *S*_BET_ could be increased from 2737 m^2^ g^−1^ to 2850 m^2^ g^−1^, while *V*_tot_ increased from 1.352 cm^3^ g^−1^ to 1.577 cm^3^ g^−1^. Up to a pore diameter *d*_P_ of 1.7 nm, the PSD behaved almost identically. Thus, it seems that the H_2_O activation had no effect on pores in this region. Pores having a larger diameter were widened by the second activation, increasing the pore diameter. The yield decreased from 9.2% to 6.4% due to the additional H_2_O activation, which was still a significantly higher total yield than could be achieved using KOH activation.

The double activated fiber was further characterized by CV and GDC. Potential windows of ±2.0 V and ±2.7 V were examined. Specific capacitances of up to 118.4 F g^−1^ derived by CV were achieved for the voltage window of ±2.0 V (Figure 6a). With increases of the voltage window, *C*_S,CV_ also increased at all scan rates. A very high value of *C*_S,CV_ = 240 F g^−1^ could be achieved at the slowest scan rate (1.0 mV s^−1^). In the GDC measurements (Figure 6b), specific capacitances of 122 F g^−1^ were obtained for discharging from 2.0 V, and 178 F g^−1^ for discharging from 2.7 V (at very low specific current of 0.025 A g^−1^). At higher currents, the sample with the higher initial voltage showed the lower specific capacitance. For the supercapacitor charged to 2.0 V, the specific power *P*_S_ calculated from the GDCs reached 4823 W kg^−1^ (Figure 6c). The maximum energy density *E*_S_ achieved was 16.5 W h kg^−1^. When charging to 2.7 V, a specific power *P*_S_ of 3640 W kg^−1^ was accomplished. The maximum specific energy *E*_S_ was 42.4 W h kg^−1^, which is extremely high compared to previously-reported energy densities [10,35,36].

## 4. Conclusions

Three different ways of obtaining activated carbon from DAHP-impregnated viscose fiber and using it as electrode material for supercapacitors were investigated. It was shown that activation by CO_2_ or H_2_O yields ACFs with a high specific surface area and a PSD that is very suitable for use as an electrode material for supercapacitors. Unlike activation with KOH, which requires immense amounts of chemicals and additional washing steps, here the fiber structure was preserved, resulting in cells with remarkably low internal resistance. The H_2_O activation also allowed the removal of the majority of heteroatoms without the need for a subsequent washing step, as shown by EDX analysis. By combining the two activation methods involving CO_2_ and H_2_O, ACFs could finally be produced. Due to their porosity with pores between 0.5 and 5.5 nm and a specific surface area of 2850 m^2^ g^−1^, morphology and purity were ideally suited for use in EDLCs. Thus, supercapacitors could be produced that exceeded a specific capacitance of 178 F g^−1^ per electrode (calculated from GDC) and an energy density of 42 W h kg^−1^ for the device.

## Figures and Tables

**Figure 1 nanomaterials-12-00677-f001:**
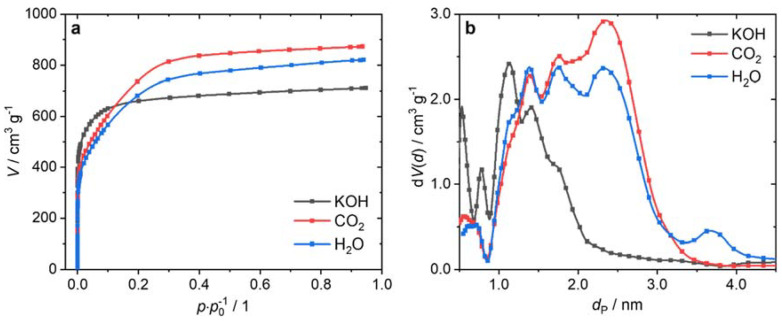
Isotherms (**a**) and PSDs (**b**) of the ACFs activated with KOH, CO_2_, and H_2_O (N_2_; 77 K; NLDFT; slit-pore model).

**Figure 2 nanomaterials-12-00677-f002:**
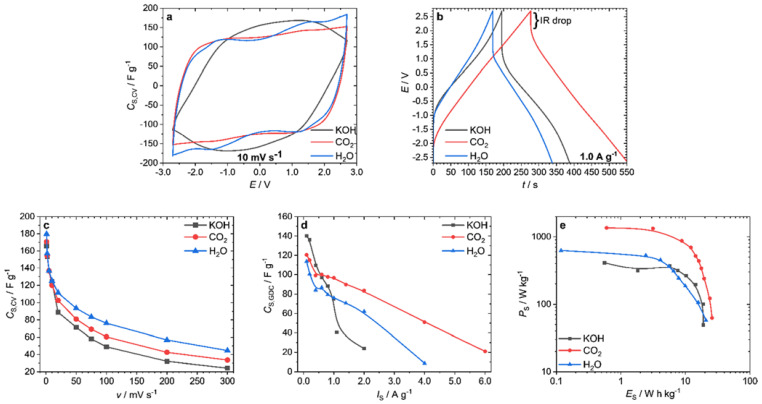
Electrochemical measurements of the EDLCs: CVs at a scan rate ν of 10 mV s^−1^ (**a**), GDC at a specific current *I*_S_ of 1.0 A g^−1^ (**b**), *C*_S,CV_ at different scan rates ν (**c**), *C*_S,GDC_ at different current densities *I*_S_ (**d**), and Ragone plot calculated from GDC per device (**e**).

**Figure 3 nanomaterials-12-00677-f003:**
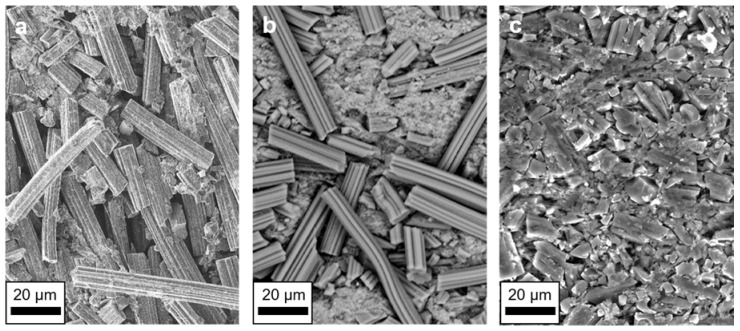
SEM micrographs of the electrode produced from the ACs activated with CO_2_ (**a**), H_2_O (**b**), and KOH (**c**).

**Figure 4 nanomaterials-12-00677-f004:**
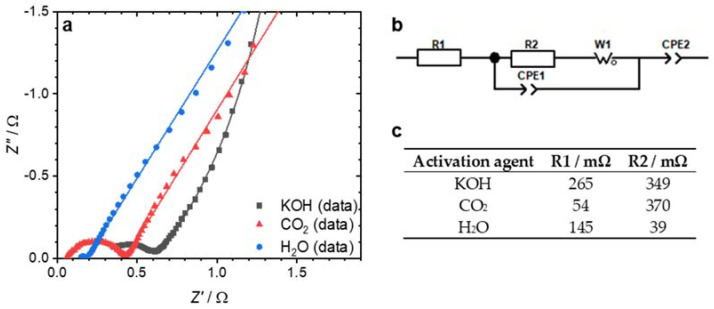
Nyquist plot of the EIS measurements (**a**). The symbols indicate the actual data measured; the lines represent the fit to the equivalent circuit, shown in (**b**). The table in (**c**) shows the values for resistors R1 and R2 determined by the fit.

**Figure 5 nanomaterials-12-00677-f005:**
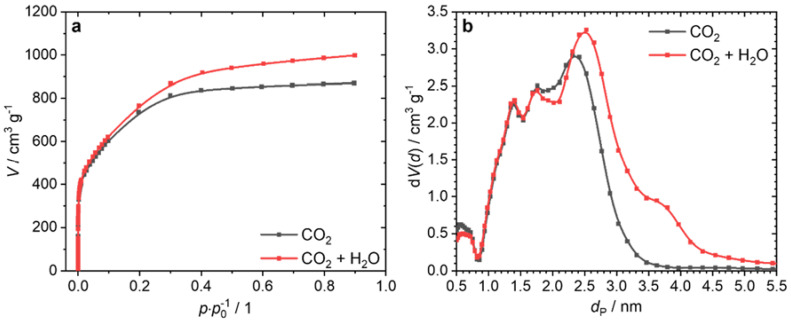
Isotherms (**a**) and PSDs (**b**) of the ACFs activated with CO_2_, and with CO_2_ followed by H_2_O (N2; 77 K; NLDFT; slit-pore model).

**Figure 6 nanomaterials-12-00677-f006:**
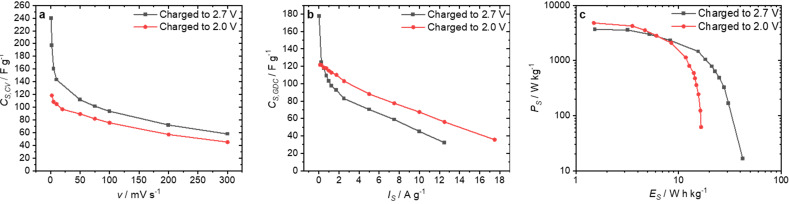
Electrochemical measurements on the ACF, which was activated twice (CO_2_ and H_2_O): *C*_S,CV_ at different scan rates ν (**a**), *C*_S,GDC_ at different current densities *I*_S_ per electrode (**b**), and Ragone plot calculated from GDC for the whole device (**c**). The values in black were obtained for charging the EDLC to 2.7 V, and those in red for charging to 2.0 V.

**Table 1 nanomaterials-12-00677-t001:** Data on the porosity (specific surface area *S*_BET_ and total pore volume *V*_tot_) and the total yield of the ACFs.

Activation Agent	*S*_BET_/m^2^ g^−1^	*V*_tot_/cm^3^ g^−1^	Total Yield/%
KOH	2553	1.10	2.1
CO_2_	2737	1.35	9.2
H_2_O	2516	1.27	23.9

**Table 2 nanomaterials-12-00677-t002:** Chemical composition of the fabricated electrodes from the ACFs. The values are calculated from at least 20 EDX spectra per sample and are arithmetic mean values with the corresponding confidence interval at a significance level of 0.05.

Activation Agent	C/at.%	O/at.%	P/at.%	N/at.%
KOH	97.4 ± 0.2	2.6 ± 0.2	0.0	0.0
CO_2_	88.3 ± 0.7	8.0 ± 0.4	2.9 ± 0.2	0.9 ± 0.2
H_2_O	98.6 ± 0.3	1.4 ± 0.3	0.0	0.0

## Data Availability

All the data is available in this manuscript.

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
