# Peer review of "Comparative Behavior of Viscose-Based Supercapacitor Electrodes Activated by KOH, H2O, and CO2"

_nanomaterials, 2022, doi:10.3390/nano12040677_

Round 1

Reviewer 1 Report

The abstract should be revised more and specific details.

The author should provide structural properties like XRD, XPS of ACFs activated with KOH, CO2, and H2O.

The author should provide more details of the electrical properties with various scan rates of best sample and compare with reported results.

In SEM micrographs, the author should provide higher magnification images and TEM.

EIS and CV curves mismatch?

Conclusions should be revised in more detail.

Reviewer 2 Report

The paper illustrates in a clear way the base principles and the procedures that the authors have employed, reaching well explained results. Only a minor revision is required. 

Line 39: it is reported that supercaps are devices "having a very high power density (< 10000 W kg-1)". Nevertheless in literature (e.g. 10.1038/srep04452) slightly higher power densities have been reported. Please, define which kind of supercaps can have a maximum power density of 10 kW kg-1 or extend the argumentation.

Line 194, 264, 267, and 269: it seems that there is an error in a reference link. Please correct.

Reviewer 3 Report

This work viscose fibers were chosen as precursor material. The potassium hydroxide, carbon dioxide and water vapor were applied and their effects on the porosity of the ACFs and their properties on the electrochemical performance of the supercapacitor were investigated. The physical activation after carbonization by means of carbon dioxide and water vapor was considered. In addition, the chemical activation by means of potassium hydroxide, which took place during the carbonization step, was examined. The assembled supercapacitors could be produced that exceeded an energy density of 42 Wh kg-1. These results demonstrate the outstanding electrochemical properties of viscose-based activated carbon fibers for use as electrode materials in energy storage devices such as supercapacitors. After reading the manuscript carefully and thoughtful consideration, I suggest to accept the manuscript after minor revision, and the comments are as follows:

  1. The background should be introduced in detail. Add the more discussions about the current research situation of biomass based carbon materials and carbon fibers (ACFs) based carbon materials in “Introduction”.
  2. For “preparation of the activated carbon fibers” section. I suggest substituting “mmol L-1” for “mmol l-1”, “L h-1” for “l h-1”. L was adopted by the CGPM in order to avoid the risk of confusion between the letter l and the number 1 (Guide for the Use of the International System of Units). The script letter l is not an approved symbol for the liter.
  3. The highest total yield of 9 %is achieved with activation using H2O, which is higher than KOH (2.1 %) and CO2 (9.2 %). Please explain the relevant mechanism.
  4. The abscissa of Figure 2cis wrong, the v means scan rates not the specific capacitance, please modify it.
  5. In Figure 2c, H2O activated sample has the best performance. But in Figure 2d, CO2activated sample has the more outstanding performance. Please explain the large difference due to different calculation methods.

Round 2

Reviewer 1 Report

I agree with the changes made and believe the manuscript can be submitted for publication in the present form.